# Assessing Consumer Preferences and Willingness to Pay for Agricultural Landscape Attributes in Lithuania

**Lucia Rocchi** [1,*] **, Anastasija Novikova** [2] **and Bernardas Vaznonis** [2]

1    Department of Agricultural, Food and Environmental Sciences, University of Perugia, 74-06121 Perugia, Italy
2    Vytautas Magnus University, K. Donelaičio Str. 58, 44248 Kaunas, Lithuania
*    Correspondence: lucia.rocchi@unipg.it

**Abstract:** Agricultural activities play an important role in shaping landscape, in particular, in Lithuania, where agricultural land accounts for more than half of the surface land. These landscape services are used by people as public goods, contributing to societal welfare. The aim of this study is to assess the value of agricultural landscape, identifying consumer preferences and willingness to pay for agricultural landscape attributes using the choice experiment (CE) method. The article analyses the agricultural landscape services and attributes the condition of which depends directly on the farming systems and practices applied. Four attributes of agricultural landscape were selected for the research: (i) scenic views; (ii) variety of flora and fauna species, (iii) recreational infrastructure and services, and (iv) objects of cultural heritage. The latent class approach (LCA) was used for analysis of heterogenous preferences among the Lithuanian residents. The findings provided quantitative information related to the demand for agricultural landscape goods, identifying a particular interest for the objects of cultural heritage, in relation to which the willingness to pay is particularly high. It is necessary to understand and analyse the preferences for agricultural landscape services with the purpose of improvement of the agricultural policy measures in order to change the farmers' performance towards sustainability. The case study offers new empirical arguments for agri-environmental policy development.

**Keywords:** agricultural landscape; choice experiments; Lithuania; preferences; public goods; willingness to pay

## 1. Introduction

According to the European Landscape Convention, landscape is described as "an area, as perceived by people, whose character is the result of the action and interaction of natural and/or human factors" [1]. Landscape is the outcome of the complex interactions between social, ecological, and geophysical processes. Physical land use and cover change are often the result of human decision making [2]. Every landscape consists of a set of components and structures: types of territories, social perceptions and constantly changing natural, social and economic forces [1].

Agriculture and the implementation of the Common Agricultural Policy measures have a significant impact on shaping the cultural landscape in the European Union. Pouta et al., 2014, highlight the differences between natural and agricultural landscapes, noting that "besides cultivated landscapes and semi-natural biotopes (such as grazing lands), agricultural landscapes are comprised of various elements, including wild nature (plants, rocks, water), man-made elements such as buildings of different ages, roads and transmission lines, and animals (wild and domesticated), as well as signs of farming (bales, fences)" [3].

Besides food, fibre, and bioenergy resource production, farmers actively participate in applying different land management strategies and provide landscape services [4]. The term "landscape services" is used to emphasize the benefits of the landscape for people and

its links with economic activity and recreation [5]. Landscape services contribute to rural vitality and economic performance, as they are the functional links between landscapes and socio-economic benefits [6]. Landscape services are described as "goods and services provided by a landscape to satisfy human needs, directly and indirectly" [7], ensuring a high quality of life–health, safety and security, recreation, and good social relations [8]. Main landscape services include provisioning services (crops, feeds, livestock, and industrial production), regulatory services (water retention, pollination, carbon storage) and cultural services (tourism, cultural heritage, housing, inspiration, hobby farming), supporting services, and biodiversity [9,10]. Most people associate landscape services with specific local landscape elements [11].

There is an increasing pressure on the agriculture to meet the demands for non-commodity outputs, in particular, multiple landscape services. Therefore, agricultural landscape features can play a key role in making the European rural areas more resilient to the major environmental challenges of the 21st century, including pollution, climate change, biodiversity loss, and reduction of water resources [12]. For better development of agricultural policy measures that affect landscape qualities, it is necessary to assess which attributes or features of the agricultural landscape are preferred by the members of the society [3]. Agricultural systems and landscapes should be valued and managed for the multiple benefits they provide [13]. The attitude of the general public towards agricultural landscape attributes and recognition of public perceptions form the basis for the development of a coherent agricultural policy. The most demanded landscape services should become one of the targets of the agricultural policy. The demanded landscape services could be ensured through the agricultural policy measures by regulating the agricultural production, consumption, and protection of natural resources [10].

Good farming practices may significantly increase these values of the landscape, while the negative effects of farming may lead to landscape degradation. Several studies [4,14,15] have been conducted proving the link between agricultural intensification (livestock density), farm size, and the decrease in landscape services, as well as the loss of some landscape elements. As the size of the farm increases, farming practices change, making a significant impact on the state of landscape elements. Therefore, the employed farming systems and practices affect the agricultural landscape services and their attributes. Aesthetic views, recreational spaces, biodiversity, and cultural heritage objects are the attributes of the agricultural landscape, the condition of which directly depends on the results of farming activities [8]. Hence, to ensure the provision of landscape services, it is necessary to maintain functional and response diversity and landscape heterogeneity [16]. Malinga [14] states that the diversification of the landscape is implemented by maintaining biological diversity, applying traditional land use methods, and reducing the scope of negative agricultural externalities [14].

Different land management decisions determining the scope of landscape services are made in view of the farmers' internal motives and as a response to external forces, for example, policy measures [4]. However, farmers do not always have economic incentives to provide landscape services because their provision increases the costs of farming, and the provision of landscape services may not be a part of their decision-making process. The efficiency of agricultural food production affects the spatial structure of agricultural land use [15], often developed at the expense of nature and important non-food ecosystem services, including the landscape ones [8,17]. Farm management decisions are presumed to be important determinants for the composition of agricultural landscapes and the services provided to society [4]. If landscape services are a priority, policy measures should be developed to encourage farmers to provide these services. The impact of agricultural policy measures on farmers' incomes and costs leads to a significant impact on farmers' choices and thus on the provision of landscape services [15].

In Europe, a variety of agricultural landscape types were created as a result of traditional farming activities and landscape diversity maintenance over the centuries. Traditional landscapes usually represent the surviving remnants of landscape heritage of the

remote past [18]. During the last century, many landscape cultural heritage objects have lost their traditional economic and social functions (infrastructure objects, farm buildings) and have been disappearing from the European agricultural landscape [12,18]. Slamova and Belčakova [18] note that typical landscapes with mixed farming structures featuring historical and cultural characteristics specific to a region have become rare or disappeared. This is due to the fact that agriculture has become oriented towards maximum profit, requiring strict specialization of agricultural production. In the Eastern EU countries, landscape changes are particularly evident due to collectivization or land consolidation [18–21]. Stokstad et al. [15] emphasize that the land consolidation processes lead to the disappearance of meadows and rows of trees that used to separate the lands of different farmers. This example shows that simplification of the landscape, by removing some landscape elements, can be profitable for farmers. Therefore, it is important to assess the importance of the elements of cultural heritage and of biodiversity for the society. As argued by Hauser et al. [4], in Western Europe, the state of biodiversity is closely related to agricultural land use. It has been observed that biodiversity has been decreasing gradually in the areas of intensive agriculture [17,22].

In order to preserve the traditional agricultural landscape, policy measures should be based on public preferences and assessment of all the benefits provided by such landscapes and of all beneficiaries. However, as emphasized by Slamova and Belčakova [18], residents of rural areas, including farmers and other inhabitants, usually underestimate the cultural, historical, and environmental values. Czúcz et al. [12], in particular, emphasize the positive environmental outcomes of traditional agricultural landscapes. Preservation of agricultural landscape elements often results in improved air quality, water quality and quantity, reduction of greenhouse gas emissions, carbon sequestration, climate change adaptation, regulation of soil erosion and soil quality, support of biodiversity, and pollination [12]. Moreover, landscapes shaped by agriculture provide benefits to many groups in society: to the landowners who benefit from the increase in land prices; those living in agricultural landscape, and those who have the pleasure and benefit thereof, but not directly possessing the land; to the visitors who enjoy agricultural landscapes for recreation and amenity, and to others who use landscape services without being physically associated with the location [14]. The non-use value of landscapes, which arise from the existence of ecosystems, should not be discarded as well.

The agri-environmental schemes of the EU are the main instruments to promote and compensate the farmers for engagement in nature and landscape conservation activities [4]. The agri-environmental and climate measures are focused on the maintenance and improvement of the agricultural landscape. Under these measures, the Lithuanian farmers receive payments for creation of agricultural landscape services. However, these measures are not always based on consumer expectations regarding agricultural activities and the landscape services they create. Furthermore, Novikova et al. [23] proved that Lithuanian residents are both concerned about the maintenance and preservation of the agricultural landscape and about the demanded agricultural landscape services.

In this study, the Lithuanian residents' priorities are revealed through the choice of hypothetical policy measures using the choice experiment (CE) method. The research findings provide quantitative information concerning the demand for agricultural landscape services, which could be useful for policy makers. Current research could also be useful for setting future direction of common agricultural policy dealing with agricultural landscape services. The paper is organized as follows: the situation of the Lithuanian agricultural landscape and methodology, selected for the present research, are presented in Section 2. Section 3 discusses the results of the empirical research; Sections 4 and 5 present the discussion in relation to the previous research and the main conclusions.

## 2. Materials and Methods

### 2.1. Case Study Area

Lithuania, with a total area of 65,300 km$^2$, is located in the centre of Europe [24]. It consists of 10 municipalities with a total population of about 2.795.7 million [1] (Figure 1). In total, 32.5% of the residents live in rural areas, 51.87% of Lithuania's surface is agricultural land, and arable land covers 45.97% of the surface area. A big part of the Lithuanian landscape is covered by forests (about 33%) [24]. The landscape of clayey plains and undulating plateaus as well as moraine hills covers more than half of Lithuania. The landscape of sandy plains, valleys, moraine ridges, and lakes is also quite common, although unequally distributed throughout the country [25].

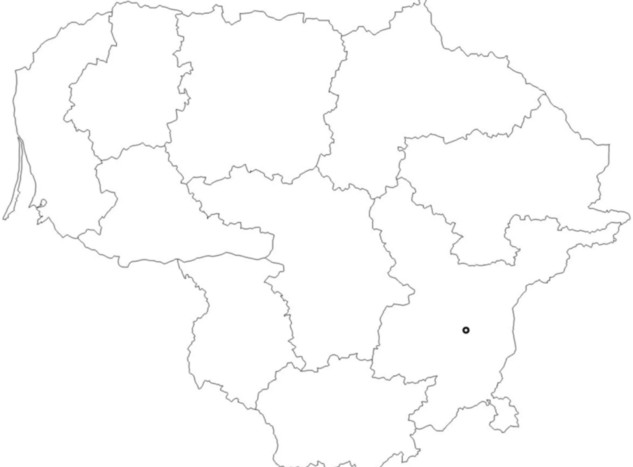

**Figure 1.** Study area. The circle highlights the capital, Vilnius.

The agriculture and forestry sector accounts for nearly 3.7% of GDP. Whereas agricultural land comprises more than half of Lithuanian territory, agricultural activities have considerable influence on the formation of the agricultural landscape and creation of the landscape amenities. Moreover, the agri-environmental measures have a big positive effect on the agricultural landscape and contribute to agri-environmental services as well as the country's National Landscape Management Plan, a state-level spatial planning document covering the country's entire territory (except for the country's territorial waters in the Baltic Sea). It sets out the principles for the protection and use of the country's landscape and the key management directions [25].

According to the SWOT analysis to evaluate the economic, social, and environmental situation in the Lithuanian agriculture and rural areas, in the period 2014–2018, organic farming, agrarian environmental, climate measures, and support to the commitments under the Natura 2000 and Water Framework Directive contributed to the protection of biodiversity, improvement of ecosystem services, and conservation of habitats and landscapes. However, although the certified organic production area increased by 1.5 times within the period 2012–2017, the variation in the number of organic farms was not uniform and has been reducing consistently since 2016 (the farms were growing in size). Organic farms accounted for a very small share in the total number of farms in the country: 1.5% in 2013 and 1.7% in 2016 [26].

According to the study by the Center for Environmental Policy [27], almost one-fourth of the agricultural area in use declared in Lithuania was involved in high nature value farming. High nature value farming is characterised by a large share of semi-natural vegetation, non-intensive farming, landscape mosaic structure, rare and protected species or their habitats, and large biodiversity. It contributes to agriculture-associated biodiversity due to the creation of more favourable conditions, as the former is directly relating to conservation and strengthening of biodiversity, agriculture-dependent habitats, and ecosystems as well as traditional rural landscapes [26]. The White Paper on Rural and Agricultural Develop-

ment of Lithuania [28] provides for the plans to support the actions that promote the high nature value farming practices that are associated with restoration and conservation of biodiversity and management of specific rural areas. It also provides for the development of special environmental measures cultivating biodiversity and landscape protection and is aimed at conservation of biodiversity and monitoring of environmental impact.

The spread of rural tourism services has been determined by the natural rural land-scapes of Lithuania that are characterised by high cultural value, protected historical, cultural values, and preserved unique rural characteristics. The number of rural tourism farmsteads and of foreign tourists staying there has been growing. In 2017, the number of rural tourism farmsteads grew by 10.4%, and the capacity at the farmsteads, by 15.0%, compared to 2012 [26].

Landscape mosaic structure is also important. Increasing the number of landscape elements could create more favourable conditions for the agriculture-associated biodiversity in Lithuania. Landscape elements provide benefit to the agricultural ecosystems and general environment and carry aesthetic and cultural value. Positive trends in the landscape mosaic structure have been identified in Lithuania in recent years. According to a study by the Center for Environmental Policy, the landscape mosaic structure was growing in the period 2014–2018. Experts at the Center for Environmental Policy share the opinion that these trends have been determined considerably by the Rural Development Policy measure "Agri-environment-climate". Nonetheless, according to the year 2015 data by the European Commission, the average number of landscape elements in Lithuania was lower than the EU's average. Moreover, the study by the Center for Environmental Policy has shown a worrying trend within the areas that are important for biodiversity protection, as 9.0% of the grasslands and similar habitats were ploughed in the period 2014–2018. In the future, it is recommended to establish territorial limitations on the provision of support in order to avoid the destruction of valuable habitats [27].

### 2.2. The Method: CE and Latent Class Modelling

Since the 1990s, the Choice Experiment (CE) has been considered the most suitable technique for non-market goods valuation [29], such as certain ecosystem services, e.g., landscape. They have the undeniable merit of enabling the measurement of both use and non-use values of non-market goods and providing a richer description of the attributes and trade-offs that individuals are willing to make, which is relevant in the public policy context [30]. In agricultural and rural planning and management, choice experiments have been increasingly applied as an instrument for policy valuation [31,32] covering a wide range of landscape typologies, attributes, and econometric specification, either alone or in combination with ecosystem services proposed by the authors [31–37].

A landscape choice experiment includes several scenarios (alternatives), each defined by a set of landscape attributes, varying on several levels. Under this approach, the choice of a specific alternative can be modeled as a function of the attributes [38]. In the present case study of the Lithuanian residents, the individuals evaluate the alternatives as a whole by trading off the attribute levels in each alternative [38]. In particular, according to the Random Utility Theory (RUT), a consumer chooses the alternative that gives the highest utility, i.e., the combination of landscape attributes that provide the highest perceived quality of landscapes. Within this theoretical framework, the subjects choose from the alternatives according to the utility function with two components, a systematic (i.e., observable) component and a random term (non-observable by the researcher) [38]. Mathematically,

$$U_{in} = V_{in}(Z_i, S_n) + \varepsilon_{in} \tag{1}$$

where

$U_{in}$ = utility provided by alternative *i* to subject *n*;
$V_{in}$ = systematic component of the utility;
$Z_i$ = vector of attributes of alternative *i*;
$S_n$ = vector of socioeconomic characteristics of respondent *n*;

$\varepsilon_{in}$ = random error term.

Several probabilistic choice models are applicable for the solution of (1); among them, the multinomial logit (MNL) model is one of the most used due to its simplicity [38–41]. According to the MNL model, the probability that individual $n$ will choose alternative $i$ ($P_{in}$) among other alternatives $j$ ($j = 1 \dots J$) of set $C_n$ (set of all the alternatives) is expressed by Equation (2) [38]:

$$P_{in} = \frac{\exp(\mu V_{in})}{\sum_{j \in C} \exp(\mu V_{jn})} \tag{2}$$

where $V_{in}/V_{jn}$ is the systematic component of the utility provided by alternatives $i$ and $j$, while $\mu$ is a scale parameter that is inversely proportional to the standard deviation of error terms and is usually assumed to be equal to one [39].

Although it has been and still is widely used, the MNL model is a basic model, limited by its assumptions [42]. Several alternative models were developed to overcome the limits and, in particular, to consider heterogeneity in taste and scale [43]. Heterogeneity is faced by two different groups of models. Mixed logit considers continuous distribution of tastes [43,44], while the Latent Class Approach (LCA) postulates their discrete distribution, which is the approach adopted in this paper. Under the LCA, the observed cases can be assigned to homogeneous preference groups, which are referred to as classes, and the respondents' heterogeneity can be represented by a finite number of classes [45,46].

For this possibility of segmentation, the LCA has been widely used in the context of consumer theory [47–49]. However, applications to ecosystem services, in general, and to landscape in particular, are numerous [34–37,46].

From a mathematical point of view, the LCA discretizes the traditional logit (see function (1)) and identifies S segments in a population, each with its own characteristics and taste [50]. The function $f(\beta)$ is discrete when $\beta$ has several finite values ($\beta_1, \beta_2, \dots \beta_n$). Therefore, the probability function (2) can be transformed to identify the probability for $n$, belonging to $s$, to choose alternative $i$, as follows:

$$P_{in} = \exp(\mu_s x_{in} \beta_s) / \sum_{j \in Cn} \exp(\mu_s x_{jn} \beta_s) \tag{3}$$

Marginal probability can be calculated as follows:

$$P_{in} = \sum_{s=1}^{S} \pi_s (\exp(\mu_s x_{in} \beta_s) / \sum_{j \in C_n} \exp(\mu_s x_{jn} \beta_s)) \tag{4}$$

When $S = \{1\}$, then $\pi_{s=1}$; therefore, the MNL specification is obtained again. $\mu_s$ is the scale parameter, and although it can vary over segments, it is usually considered equal to 1.

The identification of the number of classes is not part of the maximization process; therefore, it has to be carried out before the evaluation of parameters. It is conducted in a recurrent way, testing several models with an increasing number of classes. The optimal model is the one with the best explanatory power in order to avoid an excessive number of classes and therefore of parameters to be estimated. Usually, information criteria are applied, such as the Bayesian Information Criterion (BIC) and the Akaike Information Criterion (AIC) [51,52].

### 2.3. Survey and Questionnaire Design

Selection of the Attributes

The first stage of the CE was to identify the attributes for current research. The attributes were selected by means of an extensive literature analysis [40,41,53–56] and situation analysis of Lithuanian agricultural landscape and current agri-environmental policy [26,27,57]. Thus, four attributes for implementation of the CE (scenic views; variety of flora and fauna species; recreational infrastructure and services; objects of cultural heritage) were selected. Whereas this study specifically focuses on the attributes of agricultural landscapes, only those directly relate to agricultural management were selected.

For the choice sets to be more understandable, each attribute was illustrated by images and accompanied by the explanations about the importance of agricultural landscape maintenance and of the contribution of farming to the shaping and preservation of it. All the agricultural landscape attributes selected for the study were distributed into three levels. The lowest corresponded to status quo (no changes in agricultural landscape/no improvements). The second level corresponded to greater variety of landscape, applying extensive farming in the case of scenic views attribute, 20% improvement in variety of flora and fauna species and recreational infrastructure and services attributes, conservation of cultural and historical heritage objects for the attribute of objects of cultural heritage. The highest level (i.e., the best possible performance scenario) corresponded to mosaic landscape regulating not only the farming style, but also the land use and crop structure in the case of scenic views, 50% improvement in variety of flora and fauna species and recreational infrastructure and services attributes, increase in cultural and historical heritage objects for the attribute of objects of cultural heritage. The last attribute, monetary value, was expressed as the annual personal contribution amount to preserve the goods of agricultural landscape in EUR, proposed for a period of 5 years. Table 1 provides an overview of the selected agricultural landscape attributes and their levels of representation.

**Table 1.** Description of the representation of landscape attribute levels.

| Attribute | Description and Levels |
|---|---|
| Scenic views | - no variety of agricultural landscape<br>- greater variety of landscape, applying extensive farming<br>- mosaic landscape, regulating not only farming style, but also land use and crop structure |
| Variety of flora and fauna species | - no change<br>- enhancing of variety of flora and fauna diversity by 20<br>- enhancing of variety of flora and fauna diversity by 50% |
| Recreational infrastructure and services | - no change<br>- increase in recreational infrastructure objects by 20%<br>- increase in recreational infrastructure objects by 50% |
| Objects of cultural heritage | - decline of objects of cultural heritage<br>- conservation of cultural and historical heritage objects<br>- increase in cultural and historical heritage objects |
| Cost (Payment) | Personal contribution (EUR per year for the next 5 years)<br>0 EUR<br>12 EUR<br>24 EUR<br>48 EUR |

A pre-test (N = 45) of the constructed questionnaire was performed in 2017 spring. The results were used as priors in generating an efficient statistical design using the SAS Studio software package. Moreover, the results have been presented in detail in Novikova, Vaznonis [58]. The experimental design contained 4 blocks of questionnaires, each with 4 choice sets and 4 agricultural landscape attributes per choice set. The pretest of the questionnaire was also made to ensure the validity and readability of the questionnaire by respondents. Table 2 shows the example of a choice card from the questionnaire.

**Table 2.** Example of a choice card from the questionnaire.

| Attributes | Program A | Program B | Program C | No Program |
|---|---|---|---|---|
| Scenic views, aesthetic value | greater variety/mosaic landscape | greater variety | greater variety | |
| Variety of flora and fauna species | No changes | improvement 20% | improvement 10% | |
| Recreational infrastructure and services | No changes | No changes | No changes | |
| Objects of cultural heritage | Recovery | Decline | Decline | |
| Personal contribution (EUR per year for the next 5 years) | 48 EUR | 24 EUR | 12 EUR | 0 EUR |
| Your choice | | | | |

The questionnaire was structured in three parts. The first one focused on the opinions and awareness of the respondents about agriculture's impact on landscape. The second section presented the role of agriculture in the creation of public environmental goods with particular regards to agricultural landscape and in connection to the identified attributes. In this section are introduced also the choice sets. The third part of the questionnaire was dedicated to the collection of the respondents' socio-economic characteristics (gender, age, area of residence, size of the household, and monthly net income per capita).

## 3. Results

### 3.1. Descriptive Statistics

The main survey of the CE was carried out in the period between 2018 December and 2019 May. A total of 400 questionnaires were distributed, 275 questionnaires were filled in, and 132 questionnaires were eliminated due to incorrect completion of the survey. Several respondents selected the payment values that were too high compared to their earnings. In other words, they overestimated their financial possibilities and had to be excluded from the survey as incomplete valuations. A total of 143 valid questionnaires were obtained delivering 572 choice observations. All the respondents agreed to answer the questions of the survey, confirming that they would answer them fairly. The survey was implemented using the face-to-face method; all people that met during seminars and other events were invited to participate in present survey.

The sample included more people of young, i.e., 18–39-year, age (72.7%); half of the respondents had high education. This was supported by the interest and willingness to participate in the survey by these groups of people. Approximately 70% of the respondents were women; the average household size was 2–3 (2.9) people. About 40% of the respondents had children. More than 60% of the respondents were living in urban areas, and the monthly net income per person was 650 EUR on average.

Table 3 shows the attitudes and beliefs regarding the impacts of farming on the landscape. The respondents were asked about their opinion on the impact of agricultural environment on landscape in Lithuania. The majority (about 72%) of the respondents found the impacts of agriculture on landscape positive or very positive. About 16% of them stated that the impacts of agriculture on landscape were negative or very negative. The focus was then placed on the elements of agriculture affecting the selected attributes (i.e., scenic views, variety of flora and fauna species, recreational infrastructure and services and objects of cultural heritage). There, the respondents ranked different activities of agriculture and their impact on agricultural landscape from very small to very significant. Half of the respondents believed that the land use, crop structure and preservation of perennial meadows had the biggest positive impact on the agricultural landscape. The majority of respondents ranked coastwise management (approximately 80%), preservation of ecologically important territories (approximately 75%) and preservation of cultural objects (approximately 70%) as having the biggest positive effect on agricultural landscape.

**Table 3.** Attitudes and beliefs regarding the impacts of farming on the landscape.

| *How Do You Rate the Impacts of Agriculture on Landscape*, %? | | | | |
|---|---|---|---|---|
| **Very Bad** | **Bad** | **No Impact** | **Good** | **Very Good** |
| 1.4 | 14 | 13.3 | 56.6 | 14.7 |

| *What do you think about the impact of agricultural environment on landscape*, %? | | | | | |
|---|---|---|---|---|---|
| | Very small | Small | Average | Big | Very big |
| Land use | 0 | 8.4 | 38.5 | 40.6 | 12.6 |
| Crop structure | 0.7 | 14.7 | 35.7 | 40.6 | 8.4 |
| Farming styles and intensiveness | 0 | 9.1 | 19.6 | 48.3 | 23.1 |
| Output produced | 2.1 | 20.3 | 39.2 | 30.1 | 8.4 |
| Preservation of perennial meadows | 3.5 | 16.1 | 22.4 | 35.7 | 22.4 |
| Coastwise management | 2.1 | 7.7 | 11.2 | 28.7 | 50.3 |
| Preservation of ecologically important territories | 3.5 | 5.6 | 15.4 | 31.5 | 44.1 |
| Preservation of farming objects | 2.8 | 5.6 | 30.8 | 32.2 | 28.7 |
| Preservation of recreational objects | 3.5 | 7.7 | 25.9 | 34.3 | 28.7 |
| Preservation of cultural objects | 4.9 | 7 | 18.2 | 28 | 42 |
| Preservation of natural and historical objects | 4.9 | 8.4 | 17.5 | 26.6 | 42.7 |

*3.2. Modelling Results*

At first, the data were processed with the MNL specification for the use as benchmark, and then according to the LCA. No restrictions about the parameters were established; therefore, the results could vary across the classes. All the attributes were treated as continuous in the level with the only exception of the attribute "Objects of cultural heritage", which was discrete with two improving levels. To identify the best option, several models with two or more classes were estimated, and the one with the lowest value of the AIC, BIC and AIC/N was chosen (Table 4). The latent class probability was estimated for each class along with the *p* value. As a result, a model with two classes was assumed plausible for the data in this study. Income was the invariant individual characteristic which significantly differed between the two classes.

**Table 4.** Models fit.

| | **MNL** | **LCA** |
|---|---|---|
| McFadden Pseudo R-squared | 0.18 | 0.3 |
| AIC | 1165.9 | 1269.6 |
| AIC/N | 2.2 | 2.038 |

Table 5 reports the results of both specifications. In the MNL model, all the parameters are significant and with the expected sign, with the exceptions of "Scenic view" and "Objects of cultural heritage—improvement". Considering the LCA specification, the two classes identified differ greatly. Class I is the largest and includes 85% of the sample. In Class I, only "Scenic view" is still not significant. In class II, however, only the attribute "Objects of cultural heritage—improvement" is significant. The second group is smaller (it includes only 15% of the sample) and, according to the analysis of socioeconomic attributes, has a higher proportion of younger respondents with higher income. Moreover, the attribute of personal contribution is significant with a fairly small coefficient, indicating low sensitivity to payment. In general, the outcomes show some heterogeneity, although not very high since the best model consists of only two classes, with the first one (Class I) including the majority of the sample. This kind of a result could probably be explained by the small dimension of the sample.

**Table 5.** MNL and LCA results.

| Attributes | MNL | LCA | |
| --- | --- | --- | --- |
| | | Class I | Class II |
| | (s.e.) | (s.e.) | (s.e.) |
| Scenic views | −0.02062 | −0.02623 | 0.43573 |
| | (0.09052) | (0.10388) | (0.67048) |
| Variety of flora and fauna species | 0.01053 *** | 0.00801 * | 0.03706 |
| | (0.0039) | (0.00461) | (0.02415) |
| Recreational infrastructure and services | 0.01027 *** | 0.01379 *** | −0.04408 |
| | (0.00367) | (0.0042) | (0.03136) |
| Objects of cultural heritage—conservation | 0.57406 *** | 0.84429 *** | −31.6193 |
| | (0.21053) | (0.29294) | (0.5179D + 07) |
| Objects of cultural heritage—improvement | 0.17867 | 0.37383 * | 2.69237 ** |
| | (0.18712) | (0.22726) | (81.30559) |
| Payment | −0.08088 *** | −0.08401 *** | −018766 ** |
| | (0.00691) | (0.00753) | (0.08423) |
| Estimated latent class probabilities | | 84.7% *** | 15.3% *** |

Note: ***, **, * ==> Significance at 1%, 5%, 10% level.

Table 6 reports the WTPs for each attribute calculated using the delta method and its significance. The WTP of only one attribute ("Objects of cultural heritage—improvement") can be compared. Higher WTP is expressed in Class II, where higher income is present; however, "Objects of cultural heritage—improvement" is the only significant attribute for this class. Comparison of the other WTP values in MNL and the LCA models has shown that they are higher for "Variety of flora and fauna species" and "Recreational infrastructure and services" in the first specification and lower for both the level of the "Objects of cultural heritage".

**Table 6.** Willingness to pay.

| Attributes | MNL | LCA | |
| --- | --- | --- | --- |
| | | Class I | Class II |
| | (s.e.) | (s.e.) | (s.e.) |
| Scenic views | −0.25 | −0.24 | 5.17 |
| Variety of flora and fauna species | 0.13 *** | 0.095 * | 0.44 |
| Recreational infrastructure and services | 0.13 *** | 0.16 *** | 0.52 |
| Objects of cultural heritage—conservation | 7.10 *** | 10.05 * | −375.39 |
| Objects of cultural heritage—improvement | 2.21 | 4.45 *** | 31.96 *** |

Note: ***, * ==> Significance at 1%, 10% level.

Considering the trade-offs among the attributes, the MNL model does not show any clear preference between the attributes "Variety of flora and fauna species" and "Recreational infrastructure and services", while the lower level of the attribute "Objects of cultural heritage" (conservation) is greatly preferred over both. In Class I, the trade-off pinpoints a slight preference of "Variety of flora and fauna species" over "Recreational infrastructure and services", while for "Objects of cultural heritage", there is a preference for the lower level over the higher.

## 4. Discussion

One of the key objectives of the new CAP 2023–2027 is the preservation of landscapes and biodiversity. The topic of biodiversity attains particular focus in terms of its links to farmed landscapes and landscape features for the purpose of identification of the changes needed in the farm sector, the relevant CAP tools currently available, and the future direction. In this scenario, it is important to understand people's expectations and preferences to address the policy efficiently.

Land management and landscape in the Baltic area have largely been studied by historical [59,60] and ecological points of view [61,62] or in relation to the farming system pattern [63]. However, to the knowledge of the authors of the present paper, there have been few attempts to perform monetary assessment of landscape value [23]. According to the study by Novikova et al. [23], the willingness to pay for the preservation of the current level of landscape services was equal to EUR 23.59 per family per year and was determined by contingent valuation, which also identified the effect of certain personal characteristics such as age, gender or income. The results of the two studies are not directly comparable due to the different methods applied; however, in both cases, younger respondents showed higher WTP. This indicated that agricultural landscape management would attain greater interest in the future. In the LCA, Class II included, in particular, young people with higher income, resulting in greater WTP for the only significant attribute. The positive effect on the WTP was also observed in Novikova et al. [23].

Despite the awareness that landscape preferences are place-specific and vary across different regions [46], the coverage of the study was enlarged to include studies focusing on different areas. The findings of the present study regarding the Lithuanian residents' preferences for the attributes of agricultural landscape were found to be partly in line with previous research. In terms of the preferences regarding the variety of species, the research on the landscape attributes by Y.-H.J. Shr et al. [64] demonstrated that most people preferred richer variety in plant species. Their results are comparable with Junge et al. [65] who also noted that diversity and naturalness of landscape elements had a strong positive effect on assessment. The semi-natural landscape elements serve to maintain the aesthetic, historical and cultural value of the landscape [4]. Junge et al. [65] emphasizes the importance of aesthetic aspects in landscape management, planning and policy, despite the fact that these aspects are often neglected due to a lack of quantitative indicators. However, in the present study, the variety of flora and fauna species was not a significant attribute in Class II. Moreover, it showed little value as an attribute in the MNL model and in Class I. This result was not expected, as in Novikova et al. [23], where about 60% of the sample claimed a positive or very positive impact of biodiversity on the landscape.

Moreover, no significance of the attribute linked to the scenic view was unexpected. Findings by Hafner et al. [46] clearly indicate the key importance of point elements for the visual quality of the landscape, which ranked the highest. However, according to Kalinauskas et al. [66], agricultural areas in Lithuania show low landscape aesthetic quality (LAQ); therefore, it is possible that the Lithuanians do not perceive high scenic view as a fundamental attribute for the landscape, generating non-significance. This result can also be linked to the theories that highlight the way that landscape perception and preferences are shaped by learned behaviour and people's cultural background [67]. The positive trends in the landscape of Lithuania identified by the Center for Environmental Policy are already described in the case study; they are probably still not enough to make a change in perception and preference, pointing out a direction for future policy. Moreover, the understanding and valuing of the aesthetic values of agricultural landscape are considered to be an ongoing challenge [32,68].

The two levels of the attribute object of cultural heritage, however, performed better than a similar attribute in the previous research. For instance, Campbell [53] found that the cultural heritage attribute was significant in the Irish case study but was less preferred in comparison to the other attributes. The low number of landscape elements was also one of the weaknesses identified in the SWOT analysis of economic, social and environmental situation evaluation of the agriculture and rural areas in Lithuania. Therefore, nurturing of the landscape and preservation of biodiversity were identified as priority needs in the project of Strategic Plan 2023–2027 for Agriculture and Rural Development of Lithuania [57]. The plan provides for a range of actions under the measure Ecological systems—systems that benefit the climate, environment and animal welfare.

Agricultural practices, and in particular the traditional ones, are highly important in the creation of the conditions or stimulation of certain economic activities, such as recreation

and tourism, which potentially positively correlate to the aesthetic and environmental values of landscape [18]. Nonetheless, the authors of the present paper intentionally avoided the recreational attribute. The public support to provide landscape services is often associated with this type of service. However, the valuation given to the landscape attribute can differ when given by the local residents versus the visitors, as the local residents often state high preferences for the attributes associated with agricultural land cover, whereas the visitors prefer the attributes related to forest and natural land cover [15,65,69]. Pouta et al. [3] also found that citizens appreciated the presence of grazing animals in the landscape, although this does not increase the diversity or the naturalness of the landscape. Such differences are present also in the supply perspective. Malinga [14] revealed differences in economic activities related to the landscape services between the small-scale and large-scale farmers. Her research showed that "recreation was appreciated only by the large-scale farmers, while non-cultivated provisioning services (e.g., collection of wild foods, fire wood and building material) and some cultural services (e.g., cultural heritage, spiritual value and social relations) had higher importance for small-scale farmers". Therefore, a focus on the different perceptions of the landscape in Lithuania from a cultural and recreational point of view could be insightful, in particular, if addressed by comparison (i.e., from the citizens' versus tourists' perspective).

## 5. Conclusions

The study focused on the analysis of people's perceptions and willingness to pay for the attributes of agricultural landscape, using Lithuania as the case study, with a particular regard to the possible presence of heterogeneity. According to the agricultural characteristics of Lithuania, the agriculture sector plays an important role in the whole economy of the country. Moreover, the arable land accounts for about half of the surface; therefore, different farming practices and agri-environmental measures have a considerable impact on the landscape formation and shaping in Lithuania and contribute to the achievement of the goals of the Green Deal. The obtained research results may find practical application in the development of agricultural policy measures with the purpose to change the farmers' performance towards sustainability. The case study offers new empirical arguments for the agri-environmental policy development.

Summarizing the results of the present research, it could be stated that the majority of the respondents referred to the positive agricultural impact on the landscape. According to them, the currently used environmentally friendly policy measures (coastwise management, preservation of ecologically important territories and preservation of cultural objects) had the biggest positive effect on agricultural landscape. Econometric modelling results provided quantitative information related to the demand for agricultural landscape goods, identifying a particular interest for the objects of cultural heritage, for which willingness to pay was particularly high. It could be stated that the study identified a positive impact of the CAP agri-environmental schemes on the landscape visuality, elements and biodiversity, expected to be improved during the 2023–2027 CAP programming period.

The study has some limitations. The number of the collected questionnaires was low and decreased further upon rejection of certain questionnaires due to improper completion. Although it is possible to have a small sample in the CE in comparison to other close methodologies (e.g., contingent valuation), it may slightly affect the results for the high share of young and highly educated participants.

Moreover, the present study included a few questions to identify the familiarity with and knowledge about the agricultural landscape when the respondents were asked to rate the impacts of agriculture on landscape and requested to share their opinion about the impact of agricultural environment on the landscape. Experience from previous studies [70] shows that respondent's experience of and familiarity with a landscape are related to their preferences for the agricultural landscape. Integration of the experience and familiarity with the agricultural landscape attributes could be a possible direction for future research. Finally, the present study supported the results provided by the pioneering study on the

valuation of ecosystem and landscape services in Lithuania, identifying higher willingness to pay among the younger age groups.

**Author Contributions:** Conceptualization, B.V., A.N. and L.R.; methodology, L.R. and A.N.; software, validation, L.R.; formal analysis, B.V., A.N. and L.R.; investigation, B.V. and A.N.; resources, B.V., A.N. and L.R.; data curation, A.N. and L.R.; writing—original draft preparation, B.V., A.N. and L.R.; writing—review and editing, B.V., A.N. and L.R.; visualization, A.N.; supervision, L.R. All authors have read and agreed to the published version of the manuscript.

**Funding:** This research received no external funding.

**Institutional Review Board Statement:** Not applicable.

**Informed Consent Statement:** Not applicable.

**Data Availability Statement:** Not applicable.

**Conflicts of Interest:** The authors declare no conflict of interest.

## Note

1　https://osp.stat.gov.lt/lietuvos-gyventojai-2021/salies-gyventojai/gyventoju-skaicius-ir-sudetis.

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
