# Peer review of "Assessing Consumer Preferences and Willingness to Pay for Agricultural Landscape Attributes in Lithuania"

_land, doi:10.3390/land11101620_

Round 1

Reviewer 1 Report

The paper is interesting, but it needs to be improved further.

Main remarks:

Introduction

L69-87-Other references at international level may be added

L115-118-Try to highlight better the main contributions of your study.

2.2. The method: CE and Latent class modelling

Please present a better background, before presenting the mathematical formulation of the approach.

The model is correct and well explained.

2.3. Survey and questionnaire design

L239- Please mention the main bibliographic references consulted.

L291-It seems that the empirical implementation had several issues. Didn't the field personal was able in situ to solve these issues, when they were carrying out the survey.

L 294-Is this number of valid questionnaires statistically significant? For these types of studies it seems small. Please justify.

The discussion and conclusions are OK.

Author Response

Dear Associate Editor,

We would like to thank you and the Reviewers for your patience and thoughtful review of our paper. We also thank you for allowing us to improve our work. All comments raise interesting issues, and their inputs are very helpful for improving and completing the manuscript. We agree with almost all the comments, and we revised the manuscript accordingly. We are confident that the new version of the paper is improved, and hope that the reviewers will find our responses to their comments satisfactory. We respond below in detail to each of the reviewers’ comments.

Reviewer 1

The paper is interesting, but it needs to be improved further.

Main remarks:

Introduction

L69-87-Other references at international level may be added

Answer: we agree with the observation: the additional references are added to this part (now at lines 102-120).

L115-118-Try to highlight better the main contributions of your study.

Answer: thank you for the suggestion: the contribution of present study is highlighted at lines 151-153.

2.2. The method: CE and Latent class modelling

Please present a better background, before presenting the mathematical formulation of the approach.

The model is correct and well explained.

Answer: We agree also with this observation; therefore, we added few lines about the use of CE in landscape valuation and introduced the model in relation to it, providing references (lines 228-244).

2.3. Survey and questionnaire design

L239- Please mention the main bibliographic references consulted.

Answer: we added the missed references.

L291-It seems that the empirical implementation had several issues. Didn't the field personal was able in situ to solve these issues, when they were carrying out the survey.

Answer: the collection of questionnaires about landscape CE is sometimes affected by the difficult perception of the issue, in particular when big data companies are not involved. The personnel had instructions to provide clarification when requested but not to intervene on their own initiative, so as to avoid opportunistic behavior and bias. It is known that people may tend to give more ‘socially’ acceptable answers during face-to-face interviews than they might in other survey methods, therefore avoiding or limiting interactions is considered a good strategy. Clearly, in our data collection it didn’t work at 100%.

L 294-Is this number of valid questionnaires statistically significant? For these types of studies it seems small. Please justify.

Answer: indeed, the number of questionnaires correctly collected is small. However, the number is not too little, as the repetition of the choices allows to have a larger number of observations. Other studies have numbers close to our (among the ones we cited, van Zanten et al., 2016 include two separated data collection as small as our). We add few lines in the conclusions as ‘limitations’.

The discussion and conclusions are OK.

Reviewer 2 Report

This paper assesses the value of agricultural landscape, identifying consumer preferences and willingness to pay for agricultural landscape attributes using the choice experiment (CE) method. The author focused on four attributes of agricultural landscape and provided quantitative information related to the demand for agricultural landscape goods. The paper is reasonable and standard, and it is recommended to be published.

Author Response

Reviewer 2

This paper assesses the value of agricultural landscape, identifying consumer preferences and willingness to pay for agricultural landscape attributes using the choice experiment (CE) method. The author focused on four attributes of agricultural landscape and provided quantitative information related to the demand for agricultural landscape goods. The paper is reasonable and standard, and it is recommended to be published.

Thank you for your words!

Reviewer 3 Report

The author uses the side definition of landscape services for the purpose of the current research. It is recommended to discuss approaches to defining landscape services (including the official regulations, as soon as the author refers to the CAP measures and landscape services in the EU) and elaborate the author's approach to understanding landscape services. The same refers to the classification of landscape services - the author should critically discuss the existing approaches and select the categories most appropriate for the current study. 

The author should explore the impacts of landscape services and agricultural activities on landscapes. The paper only says that such an impact is significant, but this issue deserves more attention. What are the most demanded services the author refers to in the introduction and how do they affect landscapes?

The paper is a case study of landscape services in Lithuania, but the introduction poorly reflects the specifics of the Lithuanian case. The author mainly discusses general EU-related issues. The country-level specifics of landscape services and their impacts on landscapes must be discussed in detail to emphasize the relevance of the study and to bring a reader closer to the landscape agenda further elaborated in results and discussion. Probably, the author could rework section 2.1 by integrating some country-specific information into the introduction. A map of the territory under study (and adjacent territories) would be helpful.

Author Response

Dear Associate Editor,

We would like to thank you and the Reviewers for your patience and thoughtful review of our paper. We also thank you for allowing us to improve our work. All comments raise interesting issues, and their inputs are very helpful for improving and completing the manuscript. We agree with almost all the comments, and we revised the manuscript accordingly. We are confident that the new version of the paper is improved, and hope that the reviewers will find our responses to their comments satisfactory. We respond below in detail to each of the reviewers’ comments.

Reviewer 3

The author uses the side definition of landscape services for the purpose of the current research. It is recommended to discuss approaches to defining landscape services (including the official regulations, as soon as the author refers to the CAP measures and landscape services in the EU) and elaborate the author's approach to understanding landscape services. The same refers to the classification of landscape services - the author should critically discuss the existing approaches and select the categories most appropriate for the current study. 

Answer: We agree with this observation and, as consequence, we revised the introduction.  In particular, the landscape definition, information about agricultural landscape and main elements are added to introduction part.

The author should explore the impacts of landscape services and agricultural activities on landscapes. The paper only says that such an impact is significant, but this issue deserves more attention. What are the most demanded services the author refers to in the introduction and how do they affect landscapes?

Answer:  thank you for this valuable observation, we have added the information about the impacts of faming activities on agricultural landscape. We also highlighted the most common attributes of agricultural landscape, and how they depend from agricultural activities, policy (lines 75-88). 

The paper is a case study of landscape services in Lithuania, but the introduction poorly reflects the specifics of the Lithuanian case. The author mainly discusses general EU-related issues. The country-level specifics of landscape services and their impacts on landscapes must be discussed in detail to emphasize the relevance of the study and to bring a reader closer to the landscape agenda further elaborated in results and discussion. Probably, the author could rework section 2.1 by integrating some country-specific information into the introduction. A map of the territory under study (and adjacent territories) would be helpful.

Answer:  We tried to find and include more information about the landscape specifics, and added it to the section 2.1, as it was dedicated to presentation of study area. At the same time, we haven’t moved or added information about specifics of the Lithuanian case, as we were seeking to show the relevance of the topic on general level, and trying to modify this part was too complicated. However, we have added the map to the 2.1. section, as it was recommended.

Round 2

Reviewer 3 Report

My Round 1 recommendations have been addressed